# Prevalence of symptom exaggeration among North American independent medical evaluation examinees: A systematic review of observational studies

Andrea J. Darzi[1,2,3], Li Wang[2,3,4], John J. Riva[1], Rami Z. Morsi[5], Rana Charide[1], Rachel J. Couban[2], Samer G. Karam[1], Kian Torabiardakani[2,3], Annie Lok[3], Shanil Ebrahim[1], Sheena Bance[6], Regina Kunz[7], Gordon H. Guyatt[1], Jason W. Busse[1,2,3,4]*

1 Department of Health Research Methods, Evidence, and Impact, McMaster University, Hamilton, ON, Canada, 2 Michael G. DeGroote National Pain Centre, McMaster University, Hamilton, ON, Canada, 3 Department of Anesthesia, McMaster University, Hamilton, ON, Canada, 4 Michael G. DeGroote Institute for Pain Research and Care, McMaster University, Hamilton, ON, Canada, 5 Department of Neurology, University of Chicago, Chicago, Illinois, United States of America, 6 Centre for Addiction and Mental Health, Toronto, ON, Canada, 7 Division of Clinical Epidemiology, Evidence-based Insurance Medicine, University Hospital Basel, Basel, Switzerland

* bussejw@mcmaster.ca

## Abstract

### Background

Independent medical evaluations (IMEs) are commonly acquired to provide an assessment of impairment; however, these assessments show poor inter-rater reliability. One potential contributor is symptom exaggeration by patients, who may feel pressure to emphasize their level of impairment to qualify for incentives. This study explored the prevalence of symptom exaggeration among IME examinees in North America, which if common may represent an important consideration for improving the reliability of IMEs.

### Methods

We searched CINAHL, EMBASE, MEDLINE and PsycINFO from inception to July 08, 2024. We included observational studies that used a known-group design or multi-modal determination method. Paired reviewers independently assessed risk of bias and extracted data. We performed a random-effects model meta-analysis to estimate the overall prevalence of symptom exaggeration and explored potential subgroup effects for sex, age, education, clinical condition, and confidence in the reference standard. We used the GRADE approach to assess the certainty of evidence.

**Data availability statement:** All relevant data are within the paper and its Supporting Information files.

**Funding:** The author(s) received no specific funding for this work.

**Competing interests:** The authors have declared that no competing interests exist.

## Results

We included 44 studies with 46 cohorts and 9,794 patients. The median of the mean age was 40 (interquartile range [IQR] 38–42). Most cohorts included patients with traumatic brain injuries (n = 31, 67%) or chronic pain (n = 11, 24%). Prevalence of symptom exaggeration across studies ranged from 17% to 67%. We found low certainty evidence suggesting that studies with a greater proportion of women (≥40%) may be associated with higher rates of exaggeration (47%, 95%CI 36–58) vs. studies with a lower proportion of women (<40%) (31%, 95%CI 28–35; test of interaction p = 0.02). Possible explanations include biological differences, greater bodily awareness, or higher rates of negative affectivity. We found no significant subgroup effects for type of clinical condition, confidence in the reference standard, age, or education.

## Conclusion

Symptom exaggeration may occur in almost 50% of women and in approximately a third of men undergoing IMEs. The high prevalence of symptom exaggeration among IME attendees provides a compelling rationale for clinical evaluators to formally explore this issue. Future research should establish the reliability and validity of evaluation criteria for symptom exaggeration and develop a structured IME assessment approach.

## Background

In 2022, Statistics Canada found that 8.0 million Canadian adults reported a disability [1] and in 2020, 64.4 million Americans reported living with disability [2]. Individuals suffering from a disabling injury or illness may be eligible to receive financial compensation and services based on their level of impairment. Determinations of impairment often rely on independent medical evaluations (IMEs), which are requested by a third party, such as an insurance company or employer, and conducted by a clinician who is not part of the patient's regular medical team [3]. Underlying this process is the concern that treating clinicians may have difficulty providing impartial assessments of their patients [4,5]. Such concerns are supported by a trial that randomized 5,888 individuals in Norway to an independent assessment or usual care and found 29% of IMEs recommended less sick leave than the treating physician (68% the same, and 3% a longer duration) [6].

Despite their widespread use and far-reaching consequences, the consistency and reliability of IMEs has been challenged. The most recent systematic review found that clinical experts assessing the same patients often dissented on whether they were disabled from working (median inter-rater reliability 0.45) [7]. Although this review suggested that standardization of the assessment process may improve the reliability of IMEs, [7] two subsequent studies failed to support this hypothesis [8]. Another potential source of variability in IME assessments is symptom exaggeration [3]. IME assessors may focus too narrowly on a biomedical model to explain symptoms,

without giving sufficient attention to psychosocial and work-related factors that may influence how individuals present their symptoms [3,9].

Patients referred for IMEs often present with subjective complaints (e.g., mental illness, chronic pain) and may feel pressure to emphasize their level of impairment to qualify for wage replacement benefits, receiving time off work, or other incentives [3,10,11]. Patients' presentation may also be affected if they perceive the assessor as representing the referring agency rather than their interests. Whether or not IME assessors consider symptom exaggeration has the potential to lead to very different conclusions; however, the prevalence of exaggeration among IME attendees is uncertain and individual studies report rates as low as 17% [12] or as high as 67% [13]. Also, terminology such as exaggeration, malingering, or over-reporting are defined inconsistently across studies, making it difficult to distinguish intentional deception from psychological amplification of distress [4,14]. We undertook the first systematic review of observational studies to explore the prevalence of symptom exaggeration among IME examinees in North America.

## Methods

We conducted our systematic review in accordance with the Preferred Reporting Items for Systematic Reviews and Meta-Analyses (PRISMA) and Meta-analysis of Observational Studies in Epidemiology (MOOSE) checklists [15,16]. (See S1 and S2 Checklists in the supplemental material) We registered our protocol on the Open Science Framework (Registration DOI: https://doi.org/10.17605/OSF.IO/64V2B) [17]. After registration but prior to data analysis, we included five meta-regressions/subgroup analyses to explore variability among studies reporting the prevalence of symptom exaggeration: (1) proportion of female participants, (2) older age, (3) level of formal education, (4) clinical condition, and (5) level of confidence in the reference standard used in the approach for evaluating symptom exaggeration.

### Data sources and searches

An experienced medical librarian (RJC) developed database-specific search strategies (S1 Table) and conducted a systematic search in CINAHL, EMBASE, MEDLINE and PsycINFO, from inception through July 08, 2024. We included English, French or Spanish studies to reduce language bias. The search strategies were developed using a validation set of known relevant articles and included a combination of MeSH headings and free text key words, such as malinger* or litigation or litigant or "insufficient effort" and "independent medical examination" or "independent medical evaluation" or "disability" or "classification accuracy". We did not use any filters for our searches to maximize sensitivity. We screened the reference lists of all included studies for additional eligible articles.

### Study selection

Six reviewers screened the titles and abstracts of all retrieved citations, independently and in duplicate, and subsequently the full texts of potentially eligible studies, using standardized and pre-tested forms [18]. A third senior reviewer resolved disagreements when necessary.

Eligible studies: (i) enrolled individuals presenting for an IME in North America, (ii) in the presence of external incentive (e.g., insurance claims), and (iii) assessed the prevalence of symptom exaggeration using a known group design or multi-modal determination method [19,20]. As there is no singular reliable and valid criteria (reference standard) in the literature that is used to assess for symptom exaggeration, we included known group study designs that defined their reference standard based on criteria incorporating both clinical findings and performance on psychometric testing to classify individuals as exaggerating (within diagnostic test terminology, the target positive group), or not exaggerating (the target negative group) their symptoms [21,22].

Examples of two commonly used known group designs are the Slick, Sherman, and Iverson criteria for malingered neurocognitive dysfunction [23] and the Bianchini, Greve, & Glynn criteria for malingered pain-related disability [24]. We excluded studies that used only beyond-chance scores on symptom validity tests as an indicator of symptom

exaggeration, since beyond-chance scores are infrequent and likely to result in underestimates [25–27]. We restricted our focus to North America as there may be important differences between IMEs conducted within North America where social insurance for disability is limited and Europe where social insurance is prominent. In cases where multiple studies had population overlap, we included only the study with the larger sample size.

## Data extraction and risk of bias assessment

Teams of paired reviewers abstracted data independently and in duplicate from all eligible studies using standardized, pre-tested forms. We prefaced data abstraction with calibration exercises to optimize consistency and accuracy of extractions. For all identified studies, the reviewers abstracted the following data: name of first author, year of publication, participant demographics, referral source(s), criteria for establishing symptom exaggeration and reference standard, and the prevalence of symptom exaggeration. After completing training and calibration, pairs of reviewers independently evaluated risk of bias for each included study. They used key criteria tailored to known-group designs, which were developed and pre-tested in collaboration with research methodologists. These criteria included: (i) representativeness of the study population, (ii) validity of outcome assessment (including whether the index test was administered without knowledge of the reference standard, and confidence in the reference standard), (iii) whether those with and without symptom exaggeration were similar across age groups and education level, and (iv) loss to follow-up (≥20% was considered high risk of bias). The response options for all the above risk of bias items included "definitely yes", "probably yes", "probably no" and "definitely no". Also, we evaluated whether the criteria for establishing symptom exaggeration had been shown reliable and valid. We resolved disagreements by consensus or with the help of a third senior reviewer.

We categorized the reference standard and rated our confidence in it as either: (i) 'weak' when the study declared a known-group design, however its only criterion for identifying symptom exaggeration was below-chance performance on forced-choice symptom validity testing without any corroborating clinical observations or inconsistencies in medical records. For example, a patient with a mild ankle sprain labeled as exaggerating exclusively because they failed a below-chance forced-choice test of pain threshold, with no clinical exam or review of documented pain or functional abilities; (ii) 'moderate' where most patients exaggerating symptoms were identified by forced symptom validity testing results, but some cases could be confirmed using other credible indicators. For example, a claimant insists they cannot remember simple details of their daily routine (e.g., the route to their kitchen), yet is casually observed navigating complex tasks with no apparent cognitive difficulty; or (iii) 'strong' where exaggeration was determined by either forced symptom validity testing results or other credible clinical evidence. For example, a clinical finding that would classify a patient presenting with persistent post-concussive complaints after a very mild head injury as exaggerating symptoms would include claims of remote memory loss (e.g., loss of spelling ability).

## Data synthesis and analysis and certainty in the evidence assessment

We used a random-effects model to pool data for the prevalence of symptom exaggeration among IME examinees and a Freeman-Tukey double arcsine transformation to stabilize the variance [28,29]. This transformation avoids producing confidence intervals (CIs) that include values lower than 0% or greater than 100% [28,29]. We used the DerSimonian and Laird method [30] to pool estimates of symptom exaggeration based on the transformed values and their variances, and then the harmonic mean of sample sizes for back-transformation to the original units of proportions [31].

We assessed the certainty of evidence based on the Grading of Recommendations Assessment, Development, and Evaluation (GRADE) approach [32]. This approach considers risk of bias, indirectness, inconsistency, imprecision, and small study effects, to appraise the overall certainty of evidence as high, moderate, low, or very low [32]. We estimated that if 20% of IME attendees presented with symptom exaggeration, that would be sufficiently frequent to justify formal evaluation for exaggeration by IME evaluators. Therefore, we rated down for imprecision if the 95%CI associated with the

prevalence of symptom exaggeration included 20%. When there were at least 10 studies contributing to meta-analysis, we evaluated small study effects by visual inspection of the funnel plot for asymmetry and calculation of Egger's test [33].

## Subgroup analyses, meta-regression, and sensitivity analyses

We assessed heterogeneity across studies contributing to our pooled estimate of symptom exaggeration using both a statistical test and visual inspection of forest plots. We did not calculate $I^2$ as it can be misleading in cases where the estimates of precision are very narrow due to large sample sizes. Instead, we estimated the between-study variance with tau-squared ($\tau^2$), which provides an absolute measure of heterogeneity. We considered $\tau^2 < 0.05$ as low, between 0.05–0.1 as moderate, and >0.1 as substantial heterogeneity [34].

We assessed the variability between studies based on five hypotheses. We assumed a higher prevalence of symptom exaggeration with: (1) greater strength of the reference standard, (2) higher proportion of female participants, (3) older age, (4) lower level of formal education, and (5) higher risk of bias on a component-by-component basis. We also explored for subgroup effects based on type of clinical condition but did not pre-specify an anticipated direction of association. We conducted subgroup analyses if there were two or more studies in each subgroup, and evaluated credibility of significant subgroup effects using ICEMAN criteria [35].

We performed meta-regression to explore the relationship between the proportion of women, severity of the presenting complaint, mean age, and years of formal education, with the prevalence of symptom exaggeration. If meta-regression suggested an association, we used visual inspection of the associated scatterplot to estimate a threshold and conducted subgroup analysis. We performed all analyses using Stata software version 16.0 [36]. All comparisons were two-tailed, with a threshold P-value of 0.05.

## Ethics approval and consent to participate

We did not require ethics approval for this systematic review and meta-analysis due to our sole use of already published data.

## Systematic review update

Considering the speed at which studies exploring the prevalence of symptom exaggeration among IME attendees are published, we plan to update this review within the next five years [37].

## Results

Of 20,405 unique citations identified in our search, 44 English-language studies that reported on 46 cohorts and 9,794 patients were eligible for review. (Fig 1). None of the studies had overlapping cohorts. In S5 Table we detail the included and excluded studies with reasons at full text screening. Of the 46 cohorts, 67% (n = 31) reported on patients with traumatic brain injuries (TBI) with or without mixed neurological diseases, 24% (n = 11) on chronic pain patients, and 9% (n = 4) on other populations including toxic exposure (n = 1) [38], personal injury claimants that were not described (n = 1) [39], patients with memory impairment (n = 1) [13] and claimants reporting cognitive dysfunction following exposure to occupational and environmental substances (n = 1) [40]. In terms of criteria used to identify individuals who were exaggerating symptoms, 61% (n = 28) of cohorts relied on the Slick, Sherman, and Iverson criteria for probable malingered neurocognitive dysfunction [23], 24% (n = 11) on the Bianchini criteria [24], and 15% (n = 7) used other criteria such as those proposed by Greiffenstein, Gola, and Baker [41], Nies and Sweet [22] or Lees-Haley methods [42] (Table 1).

### Risk of bias

Of the 32% of studies that described their sampling method (14 of 44), 13 used consecutive sampling and one used random sampling methods to identify IME referrals. All studies reported minimal missing data (<5%). Most studies (n = 29, 64%) showed similar age and education characteristics between exaggerating and non-exaggerating groups. No study

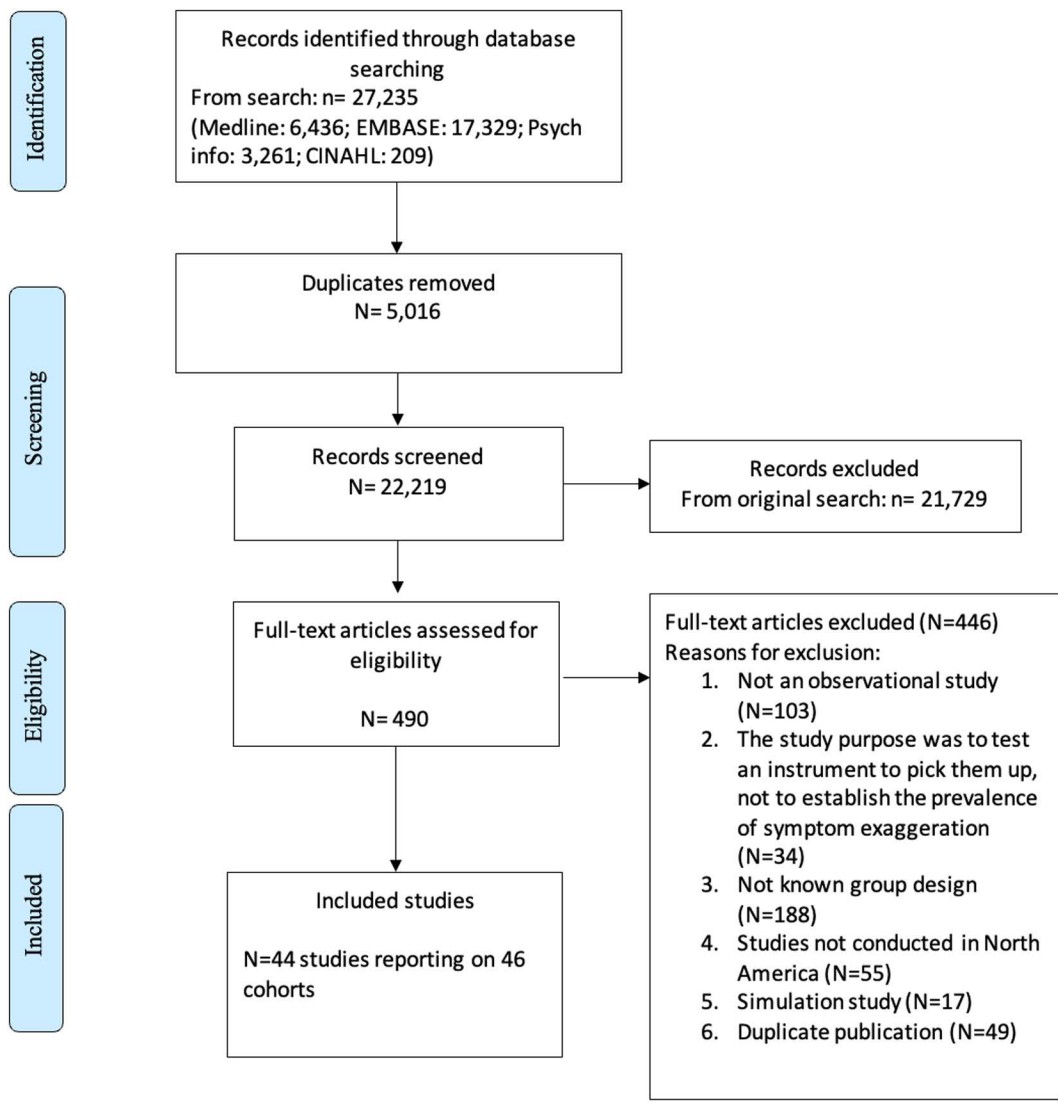

**Fig 1. PRISMA flow chart.**

explicitly stated that IME assessors administered the index test without knowledge of the reference standard. We had moderate confidence in the reference standard used by most studies (n = 35, 80%). None of the known group designs used to evaluate symptom exaggeration provided evidence of reliability and validity testing; however, there has been formal evaluation of psychometric properties of forced-choice tests that were administered in eligible studies (See S4 Table in supplementary material for details). (S2 Table).

## Prevalence of symptom exaggeration and additional analyses

The prevalence of symptom exaggeration ranged from 17% to 67%, median 33% (inter-quartile range: 25–44), and the pooled prevalence was 35% (95% confidence interval [CI]: 31–39) (low certainty evidence) (Fig 2). However, we found a significant subgroup effect, of low to moderate credibility, that studies with a higher proportion of women (≥40% vs. < 40%) may be associated with higher rates of exaggeration: 47% (95%CI 36–58) vs. 31% (95%CI 28–35) (test of interaction

**Table 1. Study Characteristics.**

| First author, year | Study Design (N) | Sampling Method | Study population | Age (mean) | % Female | Education (mean/years) | Method used to assess symptom exaggeration |
|---|---|---|---|---|---|---|---|
| **Lees-Haley, 1991 [39]** | Prospective cohort (N = 45) | NR | Personal Injury claimants | 37.8 | 58% | NR | Credibility scale: 100-item true-false scale that provides a sample of the claimant's behavior during the evaluation session and compares that behavior with both the professional experience of the clinician and with the scores of normative samples. |
| **Greiffenstein, 1995 [41]** | Retrospective cohort (N = 177) | Consecutive | TBI patients | 35.4 | NR | 12.1 | 4 Criteria: (1) improbable symptom histories, (2) improbably poor neuropsychological test scores not accounted for by physical or sensory limitations, (3) claims of subjective remote memory loss and (4) total disability in at least one major social role \* Two or more of the features had to be present. |
| **Suhr, 1997 [43]** | Retrospective cohort (N = 96) | NR | TBI patients | 35.3 | 46% | 13.2 | Criteria by Greiffenstein et al. (1994) [a] [41] with a modification that poor performance on neuropsychological tests could not be used solely to make a definitive criterion for symptom exaggeration. |
| **Costa, 1999 [13]** | Retrospective cohort (N = 42) | Consecutive | Patients with memory impairment | 40.7 | 55% | 11.5 | Criteria: (1) below chance scores on Victoria Test, (2) less than two rows on Rey-15, (3) digits forward <=3 and digits backward >=4, (4) endorsement of one or more of the following improbable procedural memory deficits: forgetting the order of walking, chewing, and swallowing movements, forgetting how to speak, constantly forgetting the way home or in own home, (5) endorsement of one or more implausible/incorrect items judged to be easy even for the moderately impaired patients with genuine amnesia, and (6) evidence of current, gainful employment \* One or more of the criteria had to be present. |
| **Van Gorp, 1999 [44]** | Retrospective cohort (N = 81) | NR | TBI patients | 36.6 | NR | 13.3 | Criteria: (1) improbable symptom history; (2) total disability in work or a major social role after 1 year from a mild closed-head injury in which loss of consciousness was less than 1 hr; (3) claims of remote or autobiographical memory loss; and (4) at least one failure on one or more neuropsychological malingering tests \* One or more of the criteria had to be present. |
| **Sweet, 2000 [45]** | Retrospective cohort (N = 63) | NR | TBI patients | 37.4 | NR | 13.8 | Criteria: (1) poor effortful performance on MDMT and/or Rey 15 Item, (2) evidence of insufficient effort on one or more traditional neuropsychological measures for which valid criteria have been established, (3) *plus* absence of credible history of neurotrauma, blatant discrepancy between potential injury and patient complaints, (4) blatant discrepancy between type of potential disorder and neuropsychological presentation, *and* (5) exaggerated patient presentation within a context of litigation or disability application \* More than one of the above criteria *and* lacking a plausible alternative explanation for patient behavior had to present |
| **Greve, 2003 [46]** | Retrospective cohort (N = 151) | NR | TBI patients | 36.6 | 34% | 12.8 | Criteria by Slick et al. (1999) [b] [23] |

*(Continued)*

**Table 1.** (Continued)

| First author, year | Study Design (N) | Sampling Method | Study population | Age (mean) | % Female | Education (mean/years) | Method used to assess symptom exaggeration |
|---|---|---|---|---|---|---|---|
| Lu, 2003 [47] | Retrospective cohort (N = 128) | Consecutive | TBI and mixed neurological conditions | 42.5 | 44% | 12.9 | Criteria by Greiffenstein et al. 1994 [41] and van Gorp et al., 1999 [44]: (1) involvement in litigation or seeking to obtain or maintain disability benefits for reported symptoms and impairments at the time of evaluation, (2) evidence of noncredible cognitive symptoms drawn from at least two of six tests designed to discreetly assess motivation and cooperation, and (3) at least one of six "external" criteria or behavioral presentations that are often observed by clinicians as signs of noncredible symptomatology<br>* All three criteria had to be present. |
| Barrash, 2004 [48] | Retrospective cohort (N = 108) | NR | TBI and mixed neurological conditions | 45.2 | 54% | 12.9 | Criteria by Greiffenstein et al. 1995 [41]: (1) Minimal brain injury (loss of consciousness and post traumatic amnesia <5min; No CT/MRI indications of brain injury and Glasgow Coma Scale>=13), (2) Clear issues of secondary gain (financial compensation, formal accommodations in work or school setting and adjudication issues), and (3) Evidence of dissimulation independent of neuropsychological performances, as indicated by at least two of the following: marked disability in a major psychosocial role, contradiction between patient and collateral sources of information, complaints of remote memory loss or other symptoms that are rarely seen as a consequence of mild head injury. |
| Heinly, 2005 [49] | Retrospective cohort (N = 344) | NR | TBI patients | 39.6 | 30% | 12.1 | Criteria by Slick et al. (1999) [b] [23] |
| Curtis, 2006 [50] | Retrospective cohort (N = 275) | NR | TBI patients | 38.7 | 28% | 12.3 | Criteria by Slick et al. (1999) [b] [23] |
| Etherton, 2006a [51] | Retrospective cohort (N = 81) | NR | Chronic pain patients | 43.3 | 36% | 11.9 | Criteria by Bianchini et al. (2005) [c] [24] |
| Greve, 2006a [52] | Not reported (N = 259) | NR | TBI patients | 38.7 | 29% | 12.5 | Criteria by Slick et al. (1999) [b] [23] |
| Greve, 2006b [53] | Not reported (N = 161) | NR | TBI patients | 39.3 | 27% | 12.3 | Criteria by Slick et al. (1999) [b] [23] |
| Greve, 2006c [54] | Not reported (N = 262) | NR | TBI patients | 38.3 | 27% | 12.2 | Criteria by Slick et al. (1999) [b] [23] |
| Greve, 2006d [40] | Retrospective cohort (N = 128) | NR | Cognitive dysfunction upon exposure to occupational and environmental substances | 40.8 | 28% | 12 | Criteria by Slick et al. (1999) [b] [23] |
| Ardolf, 2007 [55] | Retrospective cohort (N = 105) | NR | TBI and mixed neurological conditions | 40.1 | 0% | 10.5 | Criteria by Slick et al. (1999) [b] [23] |

*(Continued)*

| First author, year | Study Design (N) | Sampling Method | Study population | Age (mean) | % Female | Education (mean/years) | Method used to assess symptom exaggeration |
|---|---|---|---|---|---|---|---|
| **Greve, 2007 [56]** | Prospective cohort (N = 206) | NR | TBI patients | 39.0 | 30% | 12.6 | Criteria by Slick et al. (1999) [b] [23] |
| **Henry, 2007 [57]** | Retrospective cohort (N = 54) | NR | TBI and mixed neurological conditions | 39.8 | 46% | 14.3 | Criteria by Slick et al. (1999) [b] [23] |
| **O'Bryant, 2007 [58]** | Retrospective cohort (N = 329) | Consecutive | TBI and mixed neurological conditions | 41 | 33% | 12.7 | Criteria by Slick et al. (1999) [b] [23] |
| **Greve, 2007a [38]** | Retrospective cohort (N = 123) | NR | Toxic exposure patients | 41.3 | 29% | 12.0 | Criteria by Slick et al. (1999) [b] [23] |
| **Aguerrevere, 2008 [59]** | Retrospective cohort (N = 185) | NR | TBI patients | 37.8 | 28% | 12.4 | Criteria by Slick et al. (1999) [b] [23] |
| **Curtis, 2008 [60]** | Prospective cohort (N = 204) | NR | TBI patients | 39.6 | 29% | 12.3 | Criteria by Slick et al. (1999) [b] [23] |
| **Greve, 2008 [61]** | Prospective cohort (N = 211) | NR | TBI patients | 38.3 | 28% | 12.1 | Criteria by Slick et al. (1999) [b] [23] |
| **Ord, 2008 [62]** | Not reported (N = 93) | NR | TBI patients | 36.2 | 36% | 12.7 | Criteria by Slick et al. (1999) [b] [23] |
| **Greve, 2008b [63]** | Not reported (N TB = 109; N Chronic pain = 228) | NR | TBI and Chronic pain patients | TBI: 40.35 Chronic pain: 42.5 | TBI: 24% Chronic pain: 35% | TBI: 12.27 Chronic pain: 11.8 | Criteria by Slick et al. (1999) [b] [23] |
| **Henry, 2009 [64]** | Retrospective cohort (N = 161) | Consecutive | TBI and mixed neurological conditions | 42.0 | 41% | 13.83 | Criteria by Slick et al. (1999) [b] [23] |
| **Greve, 2009 [12]** | Retrospective cohort (N = 282) | NR | TBI patients | 37.7 | 27% | 11.7 | Criteria by Slick et al. (1999) [b] [23] |
| **Greve, 2009a [65]** | Retrospective cohort (N = 318) | Random | Chronic pain patients | 41.2 | 35% | 11.8 | Criteria by Bianchini et al. (2005) [c] [24] |
| **Greve, 2009b [66]** | Prospective cohort (N TB = 442; N Chronic pain = 378) | NR | TBI and Chronic pain patients | TBI: 38.7 Chronic pain: 42.4 | TBI: 29% Chronic pain: 37% | TBI: 12.3 Chronic pain: 11.6 | Criteria by Bianchini et al. (2005) [c] [24] |
| **Greve, 2009c [67]** | Retrospective cohort (N = 604) | Consecutive | Chronic pain patients | 42.3 | 36% | 11.7 | Criteria by Bianchini et al. (2005) [c] [24] |
| **Greve, 2009d [68]** | Retrospective cohort (N = 508) | Consecutive | Chronic pain patients | 42.1 | 35% | 11.6 | Criteria by Bianchini et al. (2005) [c] [24] |

*(Continued)*

**Table 1.** (Continued)

| First author, year | Study Design (N) | Sampling Method | Study population | Age (mean) | % Female | Education (mean/years) | Method used to assess symptom exaggeration |
|---|---|---|---|---|---|---|---|
| **Bortnik, 2010 [69]** | Retrospective cohort (N = 188) | NR | TBI and mixed neurological conditions | 42.7 | 49% | 11.9 | Criteria by Slick et al. (1999) [b] [23] |
| **Curtis, 2010 [70]** | Retrospective cohort (N = 74) | Consecutive | TBI and mixed neurological conditions | 36.3 | 35% | 13 | Criteria by Slick et al. (1999) [b] [23] |
| **Greve, 2010 [71]** | Retrospective cohort (N = 612) | Consecutive | Chronic pain patients | 41.1 | 35% | 11.7 | Criteria by Bianchini et al. (2005) [c] [24] |
| **Ord, 2010 [72]** | Retrospective cohort (N = 84) | NR | TBI patients | 39.4 | 37% | 13.0 | Criteria by Slick et al. (1999) [b] [23] |
| **Aguerrevere, 2011 [25]** | Prospective cohort (N = 108) | Consecutive | TBI patients | 39.9 | 26% | 12.4 | Criteria by Slick et al. (1999) [b] [23] |
| **Roberson, 2013 [73]** | Retrospective cohort (N = 315) | NR | TBI and mixed neurological conditions | 43.1 | 44% | 13.1 | Criteria by Slick et al. (1999) [b] [23] |
| **Bianchini, 2014 [74]** | Retrospective cohort (N = 328) | NR | Chronic pain patients | 43.3 | 35% | 12.1 | Criteria by Bianchini et al. (2005) [c] [24] |
| **Guise, 2014 [75]** | Retrospective cohort (N = 119) | Consecutive | TBI patients | 38.3 | 31% | 12.6 | Criteria by Slick et al. (1999) [b] [23] |
| **Patrick, 2014 [76]** | Retrospective cohort (N = 52) | Consecutive | TBI and mixed neurological conditions | 43.5 | 17% | 13.1 | Criteria by Slick et al. (1999) [b] [23] |
| **Aguerrevere, 2017 [77]** | Retrospective cohort (N = 348) | NR | Chronic pain patients | 43.1 | 36% | 11.7 | Criteria by Bianchini et al. (2005) [c] [24] |
| **Bianchini, 2018 [78]** | Retrospective cohort (N = 501) | Consecutive | Chronic pain patients | 42.3 | NR | 11.3 | Criteria by Bianchini et al. (2005) [c] [24] |
| **Curtis, 2019 [79]** | Retrospective cohort (N = 219) | NR | Chronic pain patients | 43.5 | 32% | 12.2 | Criteria by Bianchini et al. (2005) [c] [24] |

NR = not reported; MDMT = Medical Symptom Validity Test; PDRT = Portland Digit Recognition Test; TOMM = Test of Memory Malingering; RDS = Reliable Digit Span; FBS = Fake Bad Scale; MI = Malingering Index; WMI = Working Memory Index; PSI = Processing Speed Index

[a] Criteria by Greffeinstein et al. 1994 [41] include (1) improbable poor performance on more than two neuropsychological measures, (2) total disability in a major social role, (3) contradiction between collateral sources and symptom history, and (4) remote memory loss

[b] Criteria by Slick et al. (1999) [23] include (A) presence of substantial external incentive, (B) evidence from neuropsychological testing, (C) evidence from self-report, and (D) behaviors meeting the necessary B and C criteria are not fully accounted for by psychiatric, neurological, or developmental factors. External incentive (Criterion A) plus Criterion B and/or C evidence had to present for a diagnosis of malingering. Criterion B behaviors are sufficient for a diagnosis of malingering on their own.

[c] Criteria by Bianchini et al. (2005) [24] reflects a modification of the criteria of Slick et al. (1999) [23] and includes external incentive and meeting one of the following four conditions: (1) positive findings on either [PDRT or TOMM or RDS] and positive findings on either [FBS or MI]; (2) positive findings on [WMI and PSI] and positive findings on either [FBS or MI]; (3) positive findings on either [PDRT or TOMM] and [WMI and PSI]; or (4) significantly below chance on either [PDRT or TOMM].

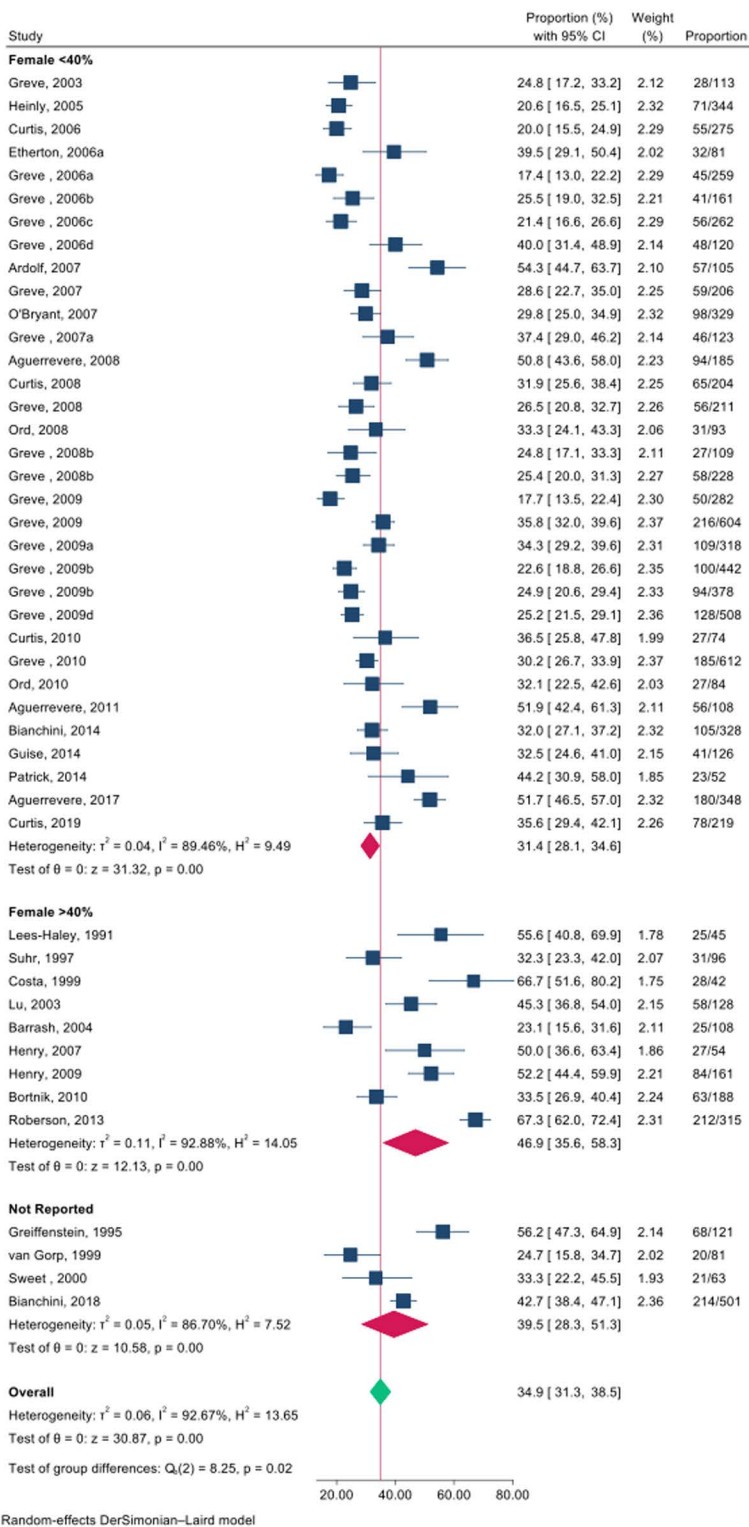

Fig 2. Forest plot for prevalence by proportion of females (P = 0.02).

**Table 2. GRADE evidence profile: prevalence of symptom exaggeration among IME attendees in North America.**

| # of studies | # of patients | Risk of bias | Inconsistency [a] | Indirectness [b] | Imprecision [c] | Publication bias | Prevalence (95% CI) | Overall certainty in the evidence |
|---|---|---|---|---|---|---|---|---|
| **Prevalence of symptom exaggeration in studies with >40% female participants** | | | | | | | | |
| 9 | 1,137 | Serious [d] | Serious [e] | Not serious | Not serious | Could not be assessed as number of included studies <10 | 46.9% (95% CI: 35.6–58.3) | Low |
| **Prevalence of symptom exaggeration in studies with <40% female participants** | | | | | | | | |
| 33 | 7,891 | Serious [d] | Serious [e] | Not serious | Not serious | Undetected; symmetric funnel plot; Egger's test P=0.16 | 31.4% (95% CI: 28.1–34.6) | Low |

[a] Inconsistency refers to variability in effect estimates across studies (i.e., heterogeneity) that could not be adequately explained.

[b] Indirectness results if the intervention, patients or outcomes are different from the research question under investigation

[c] In this review, serious imprecision is based on the position of the confidence interval relative to a 20% threshold for symptom exaggeration and if the effect on the patient, or clinical action, would differ depending on whether the upper or the lower boundary of the confidence interval represented the truth.

[d] We downgraded 1 level for risk of bias as of none the criteria used to evaluate symptom exaggeration have been formally validated.

[e] We downgraded one level for inconsistency due to a wide range in prevalence among eligible studies (17.4% to 67.3%), which was partially explained by participant sex. Specifically, studies enrolling a higher proportion of women (≥40% vs. <40%) were associated with higher rates of symptom exaggeration: 47% (95%CI 36–58) vs. 31% (95%CI 28–35; test of interaction p=0.02).

p=0.02; Fig 2, Tables 2 and S3). We did not detect any evidence of small study effects for the overall prevalence of symptom exaggeration (Egger's test P=0.13; S2 Fig) nor for the subgroup of studies with <40% women (Egger's test P=0.16; S2 Fig).

We found no significant subgroup effects for type of clinical condition (mild TBI versus chronic pain versus other conditions), confidence in the reference standard, age, or education (S3–S5 Figs). Meta-regression showed no association between prevalence of symptom exaggeration and age, level of education, or severity of presenting complaint, but did suggest an association with the proportion of female participants (S1, S6 and S7 Figs). We present all extracted data per study in S6 Table.

## Discussion

Our systematic review and meta-analysis of observational studies found low certainty evidence, rated down due to risk of bias and inconsistency, that symptom exaggeration may be common among individuals attending for IMEs in North America, affecting approximately 1 in 3 assessments. The prevalence of symptom exaggeration was higher in studies that enrolled a greater proportion of female attendees (47%) vs. a lower proportion of female attendees (31%).

### Relation to other studies

This is the first systematic review to summarize the extent of symptom exaggeration among IME attendees in North America. A previous survey of 131 US board-certified neuropsychologists conducting forensic work found that, on average, they estimated 30% of examinees claiming personal injury, disability, or workers' compensation presented with symptom exaggeration. However, estimated prevalence ranged considerably by diagnosis – from an average of 41% for mild head injuries to 2% for vascular dementia [80]. Our review found no evidence for differences in the prevalence of symptom exaggeration based on clinical condition, but most patients among studies eligible for our review presented with either mild TBI or chronic pain.

Although our review focused on IMEs in North America, data from other regions also suggest high rates of symptom exaggeration. An observational study in Spain reported that of 1,003 participants (61.5% female), drawn from unselected undergraduates, advanced psychology students, the general population, forensic psychologists, and forensic/legal medicine physicians, one-third reported having feigned symptoms or illness [81]. Data from Germany and the Netherlands

suggest that one-fifth to one-third of clients in forensic or insurance contexts exhibit symptom overreporting [82]. Further, a Swiss study found that 28% to 34% of individuals undergoing medico-legal evaluations demonstrated probable or definite symptom exaggeration [83].

Our finding suggesting that women are more likely to exaggerate symptoms vs. men is supported by a systematic review of 175 studies that found women report more bodily distress and more numerous, more intense, and more frequent somatic symptoms than men [84]. Reasons for this discrepancy are uncertain, but may include biological differences, greater bodily vigilance and awareness, and higher rates of negative affectivity vs. men [84]. When symptoms are disproportionate to objective pathology, clinicians should inquire about other factors. For example, women are more likely to experience intimate partner violence than men [85,86], and pain patients who report lifetime traumatic events experience greater pain severity [87].

Studies eligible for our review used different strategies and approaches for assessing the prevalence of symptom exaggeration. The National Academy of Neuropsychologists (NAN) and American Academy of Clinical Neuropsychology (AACN) have emphasized the use of a multimethod approach to assess symptom and performance validity. These include clinical interviews, medical records, medical investigations in certain cases, behavioural observations, and symptom and performance validity tests [88]. Specific guidance is not provided on which symptom and performance validity tests should be used, when they should be conducted, and how they should be interpreted [89].

## Strengths and limitations

Our study has several methodological strengths including (1) restricting our eligibility criteria to studies employing a known group design or multi-modal approach to assess symptom exaggeration, (2) subgroup analysis and assessment consistent with current best practices [35,90], and (3) use of the GRADE approach to evaluate the certainty of evidence.

In terms of limitations, we restricted our review to IMEs conducted in North America and eligible studies focused mainly on chronic pain and TBI. The generalizability of our findings to other jurisdictions, contexts, and clinical conditions, is uncertain. We were unable to explore the effect of cultural variability on the prevalence of symptom exaggeration as we found no studies within our inclusion criteria that addressed this issue. We did not find evidence for a subgroup effect based on confidence in the refence standard; however, there may have been insufficient variability to identify an association as almost all studies used a reference standard in which we rated moderate confidence. Another limitation of our review is the absence of a compelling reference standard for symptom exaggeration. Furthermore, even within the same reference standard, operationalization can be variable, which may affect prevalence. Another limitation of the primary studies is the lack of stratification of prevalence of symptom exaggeration according to possible effect modifiers, such as sex. Doing so would facilitate within-study subgroup analysis, which are less subject to confounding than between-study subgroup analysis. Another major limitation of the current evidence is that none of the known group approaches for evaluating symptom exaggeration have undergone reliability and validity testing.

## Implications for future research and practice

Failure to identify the contribution of symptom exaggeration towards examinee's complaints not only compromises the reliability and validity of independent assessments but may also adversely impact patient care by medicalizing psychosocial issues [91–93]. Our findings suggest that symptom exaggeration is common among patients attending for IMEs; however, we rated down the certainty of evidence due to uncertain psychometric properties of the criteria used to evaluate exaggeration. An urgent research priority is the evaluation of inter-rater reliability of known group and multi-modal systems to appraise symptom exaggeration. Validation of such assessment systems is also critical and extremely challenging, but indirect evidence of validity could be acquired by evaluating accuracy in distinguishing between volunteers who were or were not exaggerating symptoms.

Future research should investigate how cultural factors affect IME outcomes, with attention to language barriers, health beliefs, and potential biases among both examinees and assessors. Another research priority is the development and validation of a structured and comprehensive approach to identify symptom exaggeration in IME assessments. Such an approach should consider observed versus reported abilities, findings of other providers, self-reported history that is discrepant with documented history, and administration of validated tests. A further consideration for research and practice is the use of symptom validity tests that focus on malingering (e.g., Test of Memory Malingering [TOMM], Lees-Haley Fake Bad Scale [FBS]), which imply intent. Clinicians are, understandably and appropriately, hesitant to assign a label of malingering; reasons include the challenges associated with determining intent and the risk of litigation [94]. To circumvent these issues, we would suggest the use of the less value-laden term 'symptom exaggeration'.

## Conclusion

Symptom exaggeration may occur in almost 50% of women and in approximately a third of men undergoing IMEs. Assessors should evaluate symptom exaggeration when conducting IMEs using a multi-modal approach that includes both clinical findings and validated tests of performance effort, and avoid conflation with malingering which presumes intent. Priority areas for future research include establishing the reliability and validity of current evaluation criteria for symptom exaggeration, and development of a structured IME assessment approach that includes consideration of symptom exaggeration.

## Supporting information

**S1 Table. Search strategies.**
(DOCX)

**S2 Table. Risk of bias assessment.**
(DOCX)

**S3 Table. ICEMAN criteria to assess credibility of subgroup effect of female % and prevalence.**
(DOCX)

**S4 Table. Psychometric properties of tests included in symptom exaggeration criteria with list of references.**
(DOCX)

**S5 Table. Included and excluded studies at full text screening with reasons.**
(DOCX)

**S6 Table. Data extracted from included studies.**
(DOCX)

**S1 Fig. Meta-regression for proportion of females among 42 studies (p = 0.16).**
(DOCX)

**S2 Fig. a- Funnel plots of overall prevalence (Egger's test p = 0.13) and b- prevalence in subgroup of studies with female proportion <40% (Egger's test p = 0.16).**
(DOCX)

**S3 Fig. Subgroup analysis for type of conditions (test of interaction p = 0.95).**
(DOCX)

**S4 Fig. Subgroup analysis for confidence in reference standard (test of interaction p = 0.84).**
(DOCX)

**S5 Fig. Subgroup analysis for similar age and/or education between groups (test of interaction p = 0.47).**
(DOCX)

**S6 Fig. Meta-regression for average age among 46 cohorts (p = 0.18).**
(DOCX)

**S7 Fig. Meta-regression for average education level among 45 cohorts (p = 0.65).**
(DOCX)

**S1 Checklist. Preferred Reporting Items for Systematic Reviews and Meta-Analyses (PRISMA) Checklist.**
(DOCX)

**S2 Checklist. Meta-analysis of Observational Studies in Epidemiology (MOOSE) checklist.**
(DOCX)

## Acknowledgments

We would like to thank Michael Bagby from the Departments of Psychology and Psychiatry at University of Toronto for his contributions to the initial discussions around conceptualization and design of this study. No financial compensation was provided to any of these individuals.

## Author contributions

**Conceptualization:** Andrea J. Darzi, Rachel J. Couban, Regina Kunz, Gordon H. Guyatt, Jason W. Busse.

**Data curation:** Andrea J. Darzi, Li Wang, John J. Riva, Rami Z. Morsi, Rana Charide, Rachel J. Couban, Samer G. Karam, Kian Torabiardakani, Annie Lok, Shanil Ebrahim, Jason W. Busse.

**Formal analysis:** Andrea J. Darzi, Li Wang, Jason W. Busse.

**Investigation:** Andrea J. Darzi, Jason W. Busse.

**Methodology:** Andrea J. Darzi, John J. Riva, Shanil Ebrahim, Regina Kunz, Jason W. Busse.

**Project administration:** Andrea J. Darzi, Jason W. Busse.

**Resources:** Andrea J. Darzi, Jason W. Busse.

**Software:** Andrea J. Darzi, Jason W. Busse.

**Supervision:** Andrea J. Darzi, Jason W. Busse.

**Validation:** Andrea J. Darzi, John J. Riva, Sheena Bance, Regina Kunz, Gordon H. Guyatt, Jason W. Busse.

**Visualization:** Andrea J. Darzi, Jason W. Busse.

**Writing – original draft:** Andrea J. Darzi, Jason W. Busse.

**Writing – review & editing:** Andrea J. Darzi, Li Wang, John J. Riva, Rami Z. Morsi, Rana Charide, Rachel J. Couban, Samer G. Karam, Kian Torabiardakani, Annie Lok, Shanil Ebrahim, Sheena Bance, Regina Kunz, Gordon H. Guyatt, Jason W. Busse.

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
