## [Decision Letter · Decision Letter 0]

>PONE-D-24-31136>>Prevalence of Symptom Exaggeration Among North American Independent Medical Evaluation Examinees: A systematic review of observational studies>>PLOS ONE

Dear Dr. Busse,

Thank you for submitting your manuscript to PLOS ONE. After careful consideration, we feel that it has merit but does not fully meet PLOS ONE’s publication criteria as it currently stands. Therefore, we invite you to submit a revised version of the manuscript that addresses the points raised during the review process.

Thank you for your submission. I read it with enthusiasm, as it shows notable merits. Nevertheless, I noticed some concerns that I'd like the authors to address. As follows:

1) Please double-check grammar (e.g. punctuation and verb tense)

2) Please double-check refs (e.g. keep the requested style, Vancouver, and also correct formatting, punctuation and remove repetitives)

3) The Abstract is well-structured, but could benefit from conciseness. The inclusion of multiple phrases, such as "symptom exaggeration" and "observational studies," makes it somewhat repetitive. I'd suggest reducing redundant terms to improve readability;

- The logical transitions between the background, methods, and results are somewhat abrupt. For instance, it's unclear how the analysis of "symptom exaggeration" directly links to "inter-rater reliability" or whether the gender-specific findings are derived from a subset analysis or the primary outcomes. Please provide a clearer link between the methodological approach and the primary research question;

- The stats parameters (e.g. p-values, effect sizes, CIs) are provided, but more detail on the significance of these values will enhance the depth of the findings. For example, how do these rates of symptom exaggeration in IMEs impact policy, clinical practice, or future research directions? Offering these insights would broaden the reliability;

- Sample size details are provided, but the inclusion of additional information, such as variations across cohorts or specific diagnostic categories, would be useful. For instance, were patients with traumatic brain injuries or chronic pain assessed differently? This would offer a clearer understanding of subgroup analysis and generalisation;

- A brief explanation of whether the studies had high or low certainty is important;

- The abstract briefly touches on gender differences, but would benefit from elaborating on why women show higher rates of symptom exaggeration in these studies. Consider a brief sentence;

- The findings could be expanded to include more specific quantitative details, such as the range of exaggeration percentages across the cohorts or any significant moderators or predictors found in the meta-analysis;

- The conclusion is concise but could be expanded to include a brief part on how symptom exaggeration influences clinical decision-making during IMEs. Additionally, mentioning how these findings could influence future research or policy changes is essential;

- The abstract would be improved by providing the main strengths, such as prevalence rates for symptom exaggeration or differences in exaggeration across different conditions;

- The term "symptom exaggeration" is mentioned without any clear definition. Please elaborate;

4) The transition between the background on disabilities and the introduction of IMEs feels somewhat confusing. Providing a smoother transition into how symptom exaggeration affects IMEs will improve your argument;

- There are relevant stats about disabilities in North America, which effectively establishes the context. However, there is a need for a more explicit link between this background information and the objective. For example, while the IME process is mentioned, the transition to the issue of symptom exaggeration is quite abrupt

5) The Introduction touches on symptom exaggeration as a factor affecting IME outcomes. Nevertheless, there is the need to state how the authors would measure it or what variables will be examined (e.g. mental health, pain severity).

6) Explicitly state how your study addresses gaps or contradictions found in previous studies. For example, why past standardisation efforts failed. More emphasis on the relevance and novelty of the study is essential;

7) While the IME concept and symptom exaggeration are introduced, there is little detail about the methodology or variables that will be assessed;

- Please articulate your hypotheses with the rationale;

- To improve flow, consider adding a sentence explaining why IMEs are prone to variability, and how symptom exaggeration may influence;

- Summarise the objectives and hypotheses clearly in one sentence;

8) Overall, there are some worrying aspects in Introduction:

- The Introduction needs a clearer flow. Consider providing the background and context of IMEs, then move into the issue of symptom exaggeration. After that, it's important to debate what previous studies have found and highlight their gaps. You can then explain how this study will ease those. Concluding with the rationale, objectives, and hypotheses will give the Introduction a strong structure. Consider to mention why the study is important since it'll make more engaging and relevant for a broader audience;

- The text shifts abruptly between topics, like disability stats and symptom exaggeration, without clearly connecting them or stating the rationale. Brief mentioning the methods used to measure symptom exaggeration is really important;

- The authors could provide more context to strengthen their argument. IMEs often focus too narrowly on the biomedical model, overlooking psychosocial factors and work-related conditions, which influences the fairness and consistency of evaluations. The lack of standardisation and unified guidelines contributes to these issues. Additionally, biases in IME practices, such as favouring employer-provided information over patient accounts, lead to underdiagnosis and lower disability ratings compared to treating physicians. This bias fosters distrust and adds to the inconsistencies in the IME process. For more details, see 10.1097/PEC.0000000000000487 and 10.1007/978-3-319-71906-1;

- Another aspect contributing to IME variability is the difference in approaches, prognoses, and standards among evaluators. Studies show that diagnostic discordance rates can exceed 80%, with higher discordance linked to increased mortality (OR ≈ 1.2), underscoring the need for standardised protocols. Inconsistent assessments can lead to delayed treatment, denied disability benefits, and declining patient quality of life. Standardisation would improve accuracy and consistency, reducing patient burden and enhancing trust in IMEs. Furthermore, the focus on the biomedical model overlooks the role of psychological and social factors, suggesting that adopting a biopsychosocial approach could yield more accurate evaluations;

- The Introduction mentions IMEs and symptom exaggeration, but it fails to clearly state the objectives or what the study aims to explore. Providing specific objectives and hypotheses would make it easier for readers to understand the focus of the research;

- Explain how the variability in IME outcomes could be influenced by symptom exaggeration, tying it to the broader context of disability assessments;

- Strengthen the rationale by incorporating previous findings on symptom exaggeration in IMEs;

- The study focuses on symptom exaggeration as a source of variability in IMEs, an underexplored area. While past findings checked the reliability of IMEs, this work could provide important insights on how symptom exaggeration influences assessments of disability. Please consider providing a clearer explanation of why this is relevant;

- Providing more context on how symptom exaggeration might vary by condition is important; for example, variations in symptom exaggeration between conditions like traumatic brain injury and chronic pain will make the argument more cohese;

- Please clarify the relevance of the European results - what are the similarities or differences between IME in continents? And expand this part. Consider that healthcare systems are different, especially due to different geographical frameworks;

- I'd highly suggest to to introduce the concept of symptom exaggeration within the context of the study;

The Introduction is brief and lacks depth to establish a clear framework. Without the mentioned components, the Introduction feels underdeveloped and won't effectively set ground for the study. This can make the interpretation challenging without a solid rationale of the study;

9) Overall comments on Methods:

- The details about PRISMA and MOOSE are not provided. For example, the authors didn't report the refs for each checklist, and also didn't specify what workflow (e.g. screening, extraction, etc.) was followed. Including detailed information from both checklists, such as principal items, would enhance the presentation. The overall impression is, "How did the authors follow these guidelines?" Finally, why wasn't the systematic review registered on PROSPERO, the widely recommended registry for such studies?;

- The search strategy, while reported in the Sup. file, could be included in the main text to enhance clarity and robustness;

- Why include studies in other languages? Are the terms and interpretations the same across different contexts?;

- Please refine the verb tense for consistency;

- Several aspects of the design require clarification, particularly concerning the definitions, variable selection, and operationalisation. There is also a lack of explanation for the focus on specific countries. The justification for study exclusions, and the criteria for classifying symptom exaggeration could be detailed. Additionally, the handling of population overlap needs to be explained. Furthermore, if six authors conducted the screening, how was inter-rater reliability assessed? Failing to address these concerns risks making the study appear biased and unreliable;

- The eligibility criteria lack a clear rationale for including certain types of studies while excluding others. For instance, the inclusion of studies that "assessed the prevalence of symptom exaggeration using a known group design or multi-modal determination method" seems arbitrary without explaining these specific designs (c.f. 10.1212/CPJ.0000000000000092; and why they are superior or necessary for the systematic review. Please elaborate on why these aspects were chosen;

- If there is no reliable and valid standard for assessing symptoms, it's somewhat concerning to understand the reliability of the categorisation of the quality of the included studies. Actually, some studies report strategies for a reliable assessment of symptom exaggeration. Consider mentioning symptom validity tests. I think providing a detailed assessment criterion or more than one tool would be important (c.f. 10.1002/nur.10092). Consider using the Cochrane risk of bias or other tools (c.f. 10.1136/bmj.d5928; 10.1016/j.acn.2005.02.002). Also, explain how the authors ensured the reliability of the findings and what kind of psychometric assessments were performed (c.f. 10.1007/s12207-021-09436-8; 10.1016/j.acn.2005.02.002);

- The preference for including studies with larger sample size when population overlap exists introduces a potential for selection bias. Larger sample sizes do not always equate to better quality for meta-analysis. It'd be interesting to refer to relevant literature on this matter (e.g. 10.1002/sim.1186; 10.1136/bmj.d7762). I'd highly suggeest conducting robust sensitivity analyses to assess the influence of including or not certain studies (c.f. 10.1016/j.jclinepi.2004.01.<wbr style="color: rgb(34, 34, 34); font-family: Arial, Helvetica, sans-serif; font-size: small;" />018; 10.1111/rssc.12440);

10) Overall comments on Results:

- The authors could consider some approaches: a. run both fixed- and random-effects and scrutinise if there are substantial differences in the effects (i.e. if found, heterogeneity would be a really worrying aspect), b. identify studies with a high risk of bias, or small sample, remove them, and conduct sensitivity analysis (e.g. using l-o-o or cumulative sensitivity analysis). Then check the results, specifically evaluating the existence of differences in effects, CIs overlap, and changes in I² values. This would also be important when analysing subgroups;

- While beyond-chance scores are rare, they can still provide valuable insights into clinical findings (c.f. 10.1016/j.spinee.2004.11.016; 10.1186/s12874-016-0108-4). For instance, some studies have employed complementary approaches to assess symptom exaggeration (c.f. 10.1097/00001199-200004000-<wbr style="color: rgb(34, 34, 34); font-family: Arial, Helvetica, sans-serif; font-size: small;" />00006; 10.1007/BF01874896); Please elaborate;

- In studies where symptom exaggeration is being evaluated, the lack of blinding could significantly influence the results. Therefore, I would highly encourage the authors to expand this part;

- The databases used in the search are comprehensive. However, expanding the search to include other databases, such as Scopus, could ensure that no additional relevant refs are missed. Could the authors clarify why Scopus or similar databases were not included in the search strategy?;

- Additionally, more details regarding the filters applied during the search would be essential. For instance, were specific study designs or types prioritised during the screening process, and if so, which ones?;

- As recommended by Cochrane, systematic reviews should be updated to maintain their relevance and accuracy. Could the authors provide an approximate timeline for when they plan to refresh the search;

- How were duplicates handled during the search process?

- The search strategy is well-constructed and covers several important terms across databases. However, there are some concerning aspects. The use of broad terms such as 'validity' and 'disability' may result in many extraneous results, diminishing the relevance of the retrieved studies and also increasing the overall screening effort. Additionally, applying filters (e.g. specific diagnostic criteria or populations) could help enhance the reliability and focus on more relevant studies. Please clarify the rationale for not using such refinements;

- Please clarify the relevance of the transformation using F-T arcsine in your study. This is really important based on your design. The concern is to provide an explanation of parameters and the rational, enhancing clarity for readers outside the field;

- The trim-and-fill method could provide a more accurate adjustment for potential bias. Overall, after assessing any asymmetry in the funnel plot, this trims and fills missing data points to 'reach' symmetry, recalculating the overall effect size. Could the authors consider applying this approach and debating the influence on their analysis?;

- Please consider expanding the analyses by assessing Tau² (i.e. between-study variance), which could provide better insights into heterogeneity (c.f. 10.1136/bmj.327.7414.557);

- When conducting stats analyses (e.g. meta-regression, subgroup analyses), it’s essential to be cautious of small-study effects, which can influence the robustness of your findings;

- The Results are intriguing. However, the observed high variability and low certainty, as well as the association between women and higher symptom exaggeration, raise concerns about potential confounding factors. Differences in populations, diagnostic tools, and study designs likely contribute to this variability. The authors are encouraged to explore these issues more thoroughly, perhaps expanding the meta-regression to include more DVs for multivariate analysis. Consider additional approaches like structural equation modeling to reduce the number of variables, along with PCA and clustering. This would enhance the depth of your analyses, results, and overall presentation;

- Please specify test parameters (e.g. effect sizes, confidence intervals) alongside p-values for a more comprehensive understanding of the results;

- Ensure consistency in decimal places throughout the results section;

- The CIs for symptom exaggeration in women should be carefully examined, as overlapping CIs could indicate heterogeneity or residual variance;

- Given these shortcomings, I suggest the authors: a) Check for studies with small samples or wide CIs and identify possible outliers, b) Carefully consider the influence of outliers and residuals, c) Reporting I² and Tau² stats to assess heterogeneity. If heterogeneity is high, outliers could be driving the variation between studies;

11) The Discussion is well-structured but lacks conciseness, particularly in the “Relation to Other Studies” and “Implications for Future Research” sections. Streamlining these parts would enhance clarity and flow;

- While the study finds low-certainty evidence of symptom exaggeration in IMEs, the authors didn't provide sufficient arguments about what drives this low certainty. The sources of uncertainty (e.g., inconsistency, bias, imprecision) could be more clearly outlined. Also, consider debating the influence of heterogeneity;

- Some examples of how different studies operationalised symptom exaggeration differently and how this could be 'solved' seem a very interesting approach;

- Suggest practical steps for developing and validating new assessment tools for IMEs;

- Additionally, scrutinise CIs more carefully for potential overlap, as this could indicate heterogeneity;

- There is a concerning aspect regarding the connection to other findings.The arguments lack depth and focus mainly on 'correlation' with somatic symptoms, which may not fully align with IMEs or the study's objectives. Clarify the main differences between studies, focussing on the authors. The link between gender and symptom exaggeration could be simplified, and the data more sharply refined;

- Please provide a detailed explanation of how the multimethod approaches varied across studies and how the differences may have influenced the outcomes of the meta-analysis;

- The limitations are mentioned, but other biases, such as cultural or healthcare factors, could be potential confounders. Also, the lack of standardization and the absence of reliable assessments should be addressed;

- The link between Results and the Discussion could be clearer. Highlighting specific findings from the results, such as effect sizes or CIs, would enhance the reliability and make the conclusions more solid;

- The terminology distinguishing symptom exaggeration from malingering is unclear. Please clarify this distinction;

12) Tables and illustrations:

- The search strategy appears to include terms that may retrieve numerous irrelevant studies. The authors could consider refining the search by removing broad terms and avoiding overuse of wildcards. This can reduce the retrieval of unnecessary results and save future researchers from conducting an excessively lengthy and unfocused search. For example, terms like "independent" and linked terms such as "MMPI" and "work capacity" could be refined or excluded to enhance specificity. In using the current strategy, I encountered studies from adjacent fields, such as business or diagnostics for conditions unrelated to the focus of the review;

- The risk of bias Table lacks clarity and cohesion, as the studies are not organized alphabetically or by any discernible pattern, making it hard to follow. The use of consecutive sampling raises concerns about potential selection bias, which could influence the outcomes and, consequently, generalisation. Additionally, the 5% cutoff for missing data, though comprehensive, should be supported by relevant evidence to justify. While age and education are important demographics, it's unclear why other variables, such as gender, were not included. Expanding the demographic could provide a more comprehensive understanding of their influence on the findings;

- The Table on psychometric data raises concerns as it presents tests used in completely different settings, likely increasing heterogeneity. The variability in sample sizes, demographics, and the use of unreliable tools diminishes the potential for generalisation of the findings. Additionally, there are several outdated refs that should be updated to reflect current practices. Lastly, please double-check the grammar, as there are issues with punctuation and formatting that need correction;

- It's essential to note that the funnel plot is asymmetric, indicating heterogeneity. This reinforces the importance of using the trim-and-fill method to ensure robustness. The authors could also check carefully the sensitivity analysis and assess how the funnel looks after refinement. At a glance, there seem to be small-study effects, given the number of studies on left side;

- The graph on meta-regression exhibits dispersion and variability that isn't related to demographics (i.e. as indicated by the regression line). Please implement additional metrics like CIs or residuals for clarity and include important stats like the p-values, I², and other important metrics;

- The Table on study characteristics is missing important variables, like the ratio or percentage of women, including incorrect reporting (e.g. sampling being 'unclear'), and utilises confusing criteria for assessing symptom exaggeration. Please consider restructuring the Table to enhance clarity;

Overall, this is a well-conducted study, and I commend the endeavour. The findings presented are of notable importance. While this might sound lengthy, it is provided with the best intentions, aiming to align with the high standards expected in scientific communication.

While I found this study to be promising and well-conceived, I believe its current state may require extensive rounds, which could complicate the review process. To help streamline, I’ve provided some suggestions for the authors to consider. I encourage the authors to consider or check these and choose how they wish to proceed. Should they choose to resubmit after corrections, I'd be happy to read the ms again. I apologise if this isn’t the most positive news, but I see great potential in this work.

We look forward to receiving your revised manuscript.

Kind regards,

Thiago P. Fernandes, PhD

Academic Editor

PLOS ONE

Journal Requirements:

3. As required by our policy on Data Availability, please ensure your manuscript or supplementary information includes the following: 

Reviewers' comments:

Reviewer's Responses to Questions

>**Comments to the Author**

1. Is the manuscript technically sound, and do the data support the conclusions?

The manuscript must describe a technically sound piece of scientific research with data that supports the conclusions. Experiments must have been conducted rigorously, with appropriate controls, replication, and sample sizes. The conclusions must be drawn appropriately based on the data presented. >

Reviewer #1: Yes

>2. Has the statistical analysis been performed appropriately and rigorously? >

Reviewer #1: Yes

>3. Have the authors made all data underlying the findings in their manuscript fully available?

The PLOS Data policy requires authors to make all data underlying the findings described in their manuscript fully available without restriction, with rare exception (please refer to the Data Availability Statement in the manuscript PDF file). The data should be provided as part of the manuscript or its supporting information, or deposited to a public repository. For example, in addition to summary statistics, the data points behind means, medians and variance measures should be available. If there are restrictions on publicly sharing data—e.g. participant privacy or use of data from a third party—those must be specified.>

Reviewer #1: Yes

>4. Is the manuscript presented in an intelligible fashion and written in standard English?

PLOS ONE does not copyedit accepted manuscripts, so the language in submitted articles must be clear, correct, and unambiguous. Any typographical or grammatical errors should be corrected at revision, so please note any specific errors here.>

Reviewer #1: Yes

>5. Review Comments to the Author

Please use the space provided to explain your answers to the questions above. You may also include additional comments for the author, including concerns about dual publication, research ethics, or publication ethics. (Please upload your review as an attachment if it exceeds 20,000 characters)>

Reviewer #1: Dear Author(s),

I read your work with great interest, and I am pleased to congratulate you on your contribution to scientific research.

I believe that the article is novel and interesting, that it has a sufficient impact, and that it adds to the knowledge base. Plagiarism was not detected. The study appears to follow relevant guidelines and provides an original contribution to the existing scientific literature. There are no flaws in the data presented, and there are no misleading or false conclusions.

The current study is scientifically valid. The applied methodology is adequate. The reasons for performing the study are clear. I recommend the article for publication in PLOS ONE in its current form.

Sincerely,

Reviewer

>6. PLOS authors have the option to publish the peer review history of their article (what does this mean? ). If published, this will include your full peer review and any attached files.

**Do you want your identity to be public for this peer review?** For information about this choice, including consent withdrawal, please see our Privacy Policy .>

Reviewer #1: No

---

## [Author Response · Author response to Decision Letter 1]

8 Jan 2025

Academic Editor

Thank you for submitting your manuscript to PLOS ONE. After careful consideration, we feel that it has merit but does not fully meet PLOS ONE’s publication criteria as it currently stands. Therefore, we invite you to submit a revised version of the manuscript that addresses the points raised during the review process.

Reply: Thank you for your feedback, and that of the reviewers. We have considered all recommendations and comments and have addressed them in our line-by-line responses below.

General Comments:

1. Thank you for your submission. I read it with enthusiasm, as it shows notable merits. Nevertheless, I noticed some concerns that I'd like the authors to address.

Reply 1: Thank you for your review and comments. We have addressed your concerns line-by-line below.

2. Please double-check grammar (e.g. punctuation and verb tense)

Reply 2: Thank you for your comment. We have reviewed the manuscript for grammatical errors.

3. Please double-check refs (e.g. keep the requested style, Vancouver, and correct formatting, punctuation and remove repetitives)

Reply 3: Thank you for this observation. We made the appropriate revisions to our references using tracked changes in the attached manuscript.

Abstract related comments

4. The Abstract is well-structured but could benefit from conciseness. The inclusion of multiple phrases, such as "symptom exaggeration" and "observational studies," makes it somewhat repetitive. I'd suggest reducing redundant terms to improve readability.

Reply 4: Thank you for your comment. We have taken your suggestion and made our abstract more concise including the removal of redundant terms and phrases.

5. The logical transitions between the background, methods, and results are somewhat abrupt. For instance, it's unclear how the analysis of "symptom exaggeration" directly links to "inter-rater reliability" or whether the gender-specific findings are derived from a subset analysis or the primary outcomes. Please provide a clearer link between the methodological approach and the primary research question

Reply 5: We appreciate your feedback. We undertook an assessment of the prevalence of symptom exaggeration, as if it was common it may help explain the poor inter-rater reliability seen in IME assessments. By quantifying the prevalence of symptom exaggeration, clinicians and policymakers can better understand the scope of this issue and work toward improving the objectivity and reliability of IMEs. We added the following to the abstract under the introduction section:

“This study explored the prevalence of symptom exaggeration among IME examinees in North America, which if common would represent an important consideration for improving the reliability of IMEs.”

As for clarifying the methods related to the gender-specific findings we added the following in the abstract under the methods section:

“We …. explored potential subgroup effects for sex, age, education, clinical condition, and confidence in the reference standard.”

6. The stats parameters (e.g. p-values, effect sizes, CIs) are provided, but more detail on the significance of these values will enhance the depth of the findings. For example, how do these rates of symptom exaggeration in IMEs impact policy, clinical practice, or future research directions? Offering these insights would broaden the reliability.

Reply 6: Thank you. The low certainty in the findings highlighted the need for further research which we added to the conclusion of the abstract which now reads as follows:

“The high prevalence of symptom exaggeration among IME attendees provides a compelling rationale for IME evaluators to formally explore this issue. Future research should establish the reliability and validity of current evaluation criteria for symptom exaggeration and develop a structured IME assessment approach.”

7. Sample size details are provided, but the inclusion of additional information, such as variations across cohorts or specific diagnostic categories, would be useful. For instance, were patients with traumatic brain injuries or chronic pain assessed differently? This would offer a clearer understanding of subgroup analysis and generalisation.

Reply 7: Thank you. We have added clarifications, while attempting to remain concise, under the methods and results sections of the abstract. In terms of methods the added text is as noted in our response to comment 5. As for additions to the results, it now reads as follows:

“We found no significant subgroup effects for type of clinical condition, confidence in the reference standard, age, or education.”

8. A brief explanation of whether the studies had high or low certainty is important.

Reply 8: Thank you for your comment. We made clarifications to the results section by (1) describing the certainty of the body of evidence for our outcome of interest and (2) using informative statements to communicate our findings as per GRADE guidelines 26 (https://www.sciencedirect.com/science/article/pii/S0895435619304160). The language was revised in the abstract under the results section as follows:

“We found low certainty evidence suggesting that studies with a greater proportion of women (≥40% vs. <40%) may be associated with higher rates of exaggeration: 47% (95%CI 36 to 58) vs. 31% (95%CI 28 to 35; test of interaction p=0.02).”

9. The abstract briefly touches on gender differences but would benefit from elaborating on why women show higher rates of symptom exaggeration in these studies. Consider a brief sentence.

Reply 9: Thank you for this observation. We added a sentence to the Results section of the abstract as follows:

“This difference may be due to biological differences, greater bodily awareness, or higher rates of negative affectivity.”

10. The findings could be expanded to include more specific quantitative details, such as the range of exaggeration percentages across the cohorts or any significant moderators or predictors found in the meta-analysis.

Reply 10: We agree. We have revised and added more findings to our results section of the abstract as follows:

“Prevalence of symptom exaggeration across studies ranged from 17% to 67%. We found low certainty evidence suggesting that studies with a greater proportion of women (≥40% vs. <40%) may be associated with higher rates of exaggeration: 47% (95%CI 36 to 58) vs. 31% (95%CI 28 to 35; test of interaction p=0.02). We found no significant subgroup effects for type of clinical condition, confidence in the reference standard, age, or education.”

11. The conclusion is concise but could be expanded to include a brief part on how symptom exaggeration influences clinical decision-making during IMEs. Additionally, mentioning how these findings could influence future research or policy changes is essential;

Reply 11: As noted in our response to comment 6 above, we added information to the conclusion section to expand on the implications of the findings on research and practice.

12. The abstract would be improved by providing the main strengths, such as prevalence rates for symptom exaggeration or differences in exaggeration across different conditions;

Reply 12: We appreciate this feedback and have added information on the range of prevalence rates across studies and noted that we did not find a credible subgroup effect between the overall prevalence and different clinical conditions, confidence in the reference standard, age, or education. This was added to the results section of the abstract as described above in response to comment number 10.

13. The term "symptom exaggeration" is mentioned without any clear definition. Please elaborate

Reply 13: We added a short statement to the background of the abstract as follows:

“…This may be affected by symptom exaggeration where patients may feel pressure to fully convey their level of impairment to qualify for incentives.”

Introduction and objective related comments

14. The transition between the background on disabilities and the introduction of IMEs feels somewhat confusing. Providing a smoother transition into how symptom exaggeration affects IMEs will improve your argument;

- There are relevant stats about disabilities in North America, which effectively establishes the context. However, there is a need for a more explicit link between this background information and the objective. For example, while the IME process is mentioned, the transition to the issue of symptom exaggeration is quite abrupt

Reply 14: We have considered your feedback and added text (that is underlined below) to further clarify our objective. The last paragraph of our introduction now reads as follows:

“Patients referred for IMEs often present with subjective complaints (e.g., mental illness, chronic pain) and may feel pressure to emphasize their level of impairment to qualify for wage replacement benefits, receiving time off work, or other incentives (3, 9, 10). Whether or not IME assessors consider symptom exaggeration therefore has the potential to lead to very different conclusions; however, the prevalence of exaggeration among IME attendees is uncertain and individual studies report rates as low as 18% or as high as 68%. To address this gap in the literature, we undertook the first systematic review of observational studies to explore the prevalence of symptom exaggeration among IME examinees in North America.”

15. The Introduction touches on symptom exaggeration as a factor affecting IME outcomes. Nevertheless, there is the need to state how the authors would measure it or what variables will be examined (e.g. mental health, pain severity).

Reply 15: We agree and described in our methods section the types of studies and criteria used that we considered eligible in this study as noted below:

“As there is no singular reliable and valid criteria (reference standard) in the literature that is used to assess for symptom exaggeration, we included known group study designs that defined their reference standard based on criteria incorporating both clinical findings and performance on psychometric testing to classify individuals as exaggerating (within diagnostic test terminology, the target positive group), or not exaggerating (the target negative group) their symptoms (17, 18). Examples of two commonly used known group designs are the Slick, Sherman, and Iverson criteria for malingered neurocognitive dysfunction (19) and the Bianchini, Greve, & Glynn criteria for malingered pain-related disability (20).”

16. Explicitly state how your study addresses gaps or contradictions found in previous studies. For example, why past standardisation efforts failed. More emphasis on the relevance and novelty of the study is essential.

Reply 16: We undertook the first systematic review of symptom exaggeration among IME attendees. This has been explicitly stated and clarified in our introduction.

17. While the IME concept and symptom exaggeration are introduced, there is little detail about the methodology or variables that will be assessed;

- Please articulate your hypotheses with the rationale;

- To improve flow, consider adding a sentence explaining why IMEs are prone to variability, and how symptom exaggeration may influence;

- Summarise the objectives and hypotheses clearly in one sentence.

Reply 17: Thank you for your feedback. As noted in the reply to comments 14 we have now added the below to further clarify our objective and reasons for it:

“Whether or not IME assessors consider symptom exaggeration therefore has the potential to lead to very different conclusions; however, the prevalence of exaggeration among IME attendees is uncertain and individual studies report rates as low as 18% or as high as 68%.”

18. Overall, there are some worrying aspects in Introduction:

The Introduction needs a clearer flow. Consider providing the background and context of IMEs, then move into the issue of symptom exaggeration. After that, it's important to debate what previous studies have found and highlight their gaps. You can then explain how this study will ease those. Concluding with the rationale, objectives, and hypotheses will give the Introduction a strong structure. Consider to mention why the study is important since it'll make more engaging and relevant for a broader audience

Reply 18: Thank you for your feedback. We have made changes accordingly to ensure clarity and better flow as noted in our responses to comments 14 and 17.

19. The text shifts abruptly between topics, like disability stats and symptom exaggeration, without clearly connecting them or stating the rationale. Brief mentioning the methods used to measure symptom exaggeration is really important

Reply 19: The methods for measuring symptom exaggeration that we considered eligible are defined in the methods section under eligibility criteria as noted in our response to comment 15.

20. The authors could provide more context to strengthen their argument. IMEs often focus too narrowly on the biomedical model, overlooking psychosocial factors and work-related conditions, which influences the fairness and consistency of evaluations. The lack of standardisation and unified guidelines contributes to these issues. Additionally, biases in IME practices, such as favouring employer-provided information over patient accounts, lead to underdiagnosis and lower disability ratings compared to treating physicians. This bias fosters distrust and adds to the inconsistencies in the IME process. For more details, see 10.1097/PEC.0000000000000487 and 10.1007/978-3-319-71906-1

Reply 20: Thanks for this citation, which we have now included the justify the following added statement:

“Independent evaluators may focus too narrowly on a biomedical model to explain symptoms, without sufficient attention to psychosocial and work-related factors.”

21. Another aspect contributing to IME variability is the difference in approaches, prognoses, and standards among evaluators. Studies show that diagnostic discordance rates can exceed 80%, with higher discordance linked to increased mortality (OR ≈ 1.2), underscoring the need for standardised protocols. Inconsistent assessments can lead to delayed treatment, denied disability benefits, and declining patient quality of life. Standardisation would improve accuracy and consistency, reducing patient burden and enhancing trust in IMEs. Furthermore, the focus on the biomedical model overlooks the role of psychological and social factors, suggesting that adopting a biopsychosocial approach could yield more accurate evaluations

Reply 21: In terms of the need for standardised protocols, as noted in our introduction, a recent systematic review suggested that standardization of the assessment process may improve the reliability of IMEs; however, two subsequent studies have failed to support this hypothesis (Reference: Kunz R, von Allmen DY, Marelli R, Hoffmann-Richter U, Jeger J, Mager R, et al. The reproducibility of psychiatric evaluations of work disability: two reliability and agreement studies. BMC psychiatry. 2019;19(1):1-15.). This is what led us to explore another potential source of variability in IME assessments which was the prevalence of symptom exaggeration.

22. The Introduction mentions IMEs and symptom exaggeration, but it fails to clearly state the objectives or what the study aims to explore. Providing specific objectives and hypotheses would make it easier for readers to understand the focus of the research

Reply 22: We have clarified this and added to the text as noted in our responses to comments 14, 17 and 18.

23. Explain how the variability in IME outcomes could be influenced by symptom exaggeration, tying it to the broader context of disability assessments

Reply 23: Thank you for your feedback. As noted in the reply to comments 14 and 17 we have now added the text underlined below to further clarify our objective and reasons for it:

“Another potential source of variability in IME assessments is symptom exaggeration (3). Patients referred for IMEs often present with subjective complaints (e.g., mental illness, chronic pain) and may feel pressure to emphasize their level of impairment to qualify for wage replacement benefits, receiving time off wo

---

## [Decision Letter · Decision Letter 1]

>PONE-D-24-31136R1>>Prevalence of Symptom Exaggeration Among North American Independent Medical Evaluation Examinees: A systematic review of observational studies>>PLOS ONE

Dear Dr. Busse,

Thank you for submitting your manuscript to PLOS ONE. After careful consideration, we feel that it has merit but does not fully meet PLOS ONE’s publication criteria as it currently stands. Therefore, we invite you to submit a revised version of the manuscript that addresses the points raised during the review process.

Thank you for your valuable submission.

That said, there are two main concerns in this round.

Before all, I’d like to mention that I read with enthusiasm and think the study has merits to reach a broad audience. This is the main - and most important - aspect to consider in this lengthy round. As noticed while re-reading (and as pointed out by the reviewer), many of the previously raised concerns remain unaddressed.

It appears that certain important adjustments were not fully considered. The reviewer's comments highlight the need for a more detailed rebuttal, particularly clarifying and addressing all the raised concerns. Although lengthy rounds are not common at this point, I trust that with continued effort, the ms has potential and can be refined to meet the required standards.

I understand this is not the best news, but in its current form, the ms cannot proceed. The concerns raised require considerable adjustments and further work. If the authors are willing to address these issues, I would be happy to reassess the ms after the necessary changes are made.

We look forward to receiving your revised manuscript.

Kind regards,

Thiago P. Fernandes, PhD

Academic Editor

PLOS ONE

Reviewers' comments:

Reviewer's Responses to Questions

>**Comments to the Author**

1. If the authors have adequately addressed your comments raised in a previous round of review and you feel that this manuscript is now acceptable for publication, you may indicate that here to bypass the “Comments to the Author” section, enter your conflict of interest statement in the “Confidential to Editor” section, and submit your "Accept" recommendation.>

Reviewer #2: (No Response)

Reviewer #3: (No Response)

>2. Is the manuscript technically sound, and do the data support the conclusions?

The manuscript must describe a technically sound piece of scientific research with data that supports the conclusions. Experiments must have been conducted rigorously, with appropriate controls, replication, and sample sizes. The conclusions must be drawn appropriately based on the data presented. >

Reviewer #2: Partly

Reviewer #3: Yes

>3. Has the statistical analysis been performed appropriately and rigorously? >

Reviewer #2: Yes

Reviewer #3: Yes

>4. Have the authors made all data underlying the findings in their manuscript fully available?

The PLOS Data policy requires authors to make all data underlying the findings described in their manuscript fully available without restriction, with rare exception (please refer to the Data Availability Statement in the manuscript PDF file). The data should be provided as part of the manuscript or its supporting information, or deposited to a public repository. For example, in addition to summary statistics, the data points behind means, medians and variance measures should be available. If there are restrictions on publicly sharing data—e.g. participant privacy or use of data from a third party—those must be specified.>

Reviewer #2: No

Reviewer #3: Yes

>5. Is the manuscript presented in an intelligible fashion and written in standard English?

PLOS ONE does not copyedit accepted manuscripts, so the language in submitted articles must be clear, correct, and unambiguous. Any typographical or grammatical errors should be corrected at revision, so please note any specific errors here.>

Reviewer #2: Yes

Reviewer #3: Yes

>6. Review Comments to the Author

Please use the space provided to explain your answers to the questions above. You may also include additional comments for the author, including concerns about dual publication, research ethics, or publication ethics. (Please upload your review as an attachment if it exceeds 20,000 characters)>

Reviewer #2: The current manuscript is a revision about the prevalence of symptom exaggeration among IME examinees. I did not review the initial version of this manuscript, but reviewed the revision. I have checked my comments with the initial review comments, please see below.

Major:

Symptom overreporting may have multiple causes (see Merckelbach et al., 2019). Symptom exaggeration due to external incentives is related to malingering. Your reply to point 64 states: "We noticed that most studies use the terms interchangeably and often label their criteria as criteria to assess malingering. However, when using the term malingering one presumes intent. Thus we felt it was an important suggestion to note in our discussion that considering how ‘clinicians are, understandably and appropriately, hesitant to assign a label of malingering…. (45)’ the term symptom exaggeration may be a more appropriate replacement." Indeed, malingering equals symptom overreporting, but symptom overreporting does not necessarily mean malingering. In the context of IME, malingering might be more obvious to use for the manuscript, though in clinical practice one should not always assume but rather take malingering into account.

Both the introduction and discussion are very short, relating to the point 29 raised in the previous round. In my opinion, these comments are not sufficiently dealt with. Also the transitions between different topics within the Introduction and Discussion could be improved.

It is stated that French and Spanish studies were included to reduce language bias, but what about culture bias? You state this later on, but this is something that could be taken into account at an earlier level.

Risk of bias assessment: also relating to point 36> why were no tools used to assess risk of bias? E.g., Cochrane's? Also, what was the measure of agreement (e.g., Kappa)? The selection seems very subjective.

Table 2: what do the levels of inconsistency mean, especially 'serious inconsistency'? How does this relate to your conclusions?

It is concluded that the review found no evidence for differences in the prevalence of symptom exaggeration based on clinical condition, but most patients among studies eligible for our review presented with either mild TBI or chronic pain'. This was not a (preregistered) research question, and how to conclude this based on only two clinical conditions? Other clinical diagnoses might have symptom exaggeration, but are not included in the current review.

Minor:

Some reasoning errors in conclusions; if women report higher rates of symptom exaggeration, then it is logically that studies that included more women show higher raters.

There are still some punctuation errors (extra comma use, delete space) throughout the manuscript. At times the phrasing (e.g., p.23 line 72 ('Such concerns...) could be improved.

For research transparency, what were the exact key words? I know they are stated in supplemental materials and that the apparently take 3 pages to explain, but I do believe these could be stated clearly in the manuscript.

Reviewer #3: Dear authors of the article, I want to thank you for a well-prepared article.

The undoubted advantages of the article are:

The text of the article is logically structured and understandable to the reader. The language meets the level and requirements of the scientific language. The article uses modern data analysis, adequate to the purpose of the study. The research is of high importance for science and practice. The authors openly and adequately note in the article limitations, implications for future research and practice.

I certainly recommend this article for publication in the journal, but I have some questions and recommendations for improvement.:

1. why were only these databases CINAHL, EMBASE, MEDLINE and PsycINFO used for analysis?

2. In the discussion section, it is necessary to go beyond America and compare scientific data with other countries and continents

3. At the end of the paragraph "Eligible studies", clarify what does "large sample size" mean (what is the sample size in numbers)?

4. In the paragraph "We rated confidence in the reference standard as either..." describe in more detail (if possible, give examples) what ‘weak’, ‘moderate’ or ‘strong’ mean?

>7. PLOS authors have the option to publish the peer review history of their article (what does this mean? ). If published, this will include your full peer review and any attached files.

**Do you want your identity to be public for this peer review?** For information about this choice, including consent withdrawal, please see our Privacy Policy .>

Reviewer #2: **Yes: ** Sanne Houben

Reviewer #3: **Yes: ** Olga Terekhina

---

## [Author Response · Author response to Decision Letter 2]

2 Apr 2025

Reviewer 2 Comments

1. Symptom overreporting may have multiple causes (see Merckelbach et al., 2019). Symptom exaggeration due to external incentives is related to malingering. Your reply to point 64 states: "We noticed that most studies use the terms interchangeably and often label their criteria as criteria to assess malingering. However, when using the term malingering one presumes intent. Thus we felt it was an important suggestion to note in our discussion that considering how ‘clinicians are, understandably and appropriately, hesitant to assign a label of malingering…. (45)’ the term symptom exaggeration may be a more appropriate replacement." Indeed, malingering equals symptom overreporting, but symptom overreporting does not necessarily mean malingering. In the context of IME, malingering might be more obvious to use for the manuscript, though in clinical practice one should not always assume but rather take malingering into account.

Reply 2: Thank you for raising this important point regarding the distinction between malingering and symptom exaggeration or overreporting. In our review, we avoided the term malingering because it presupposes evidence of intent, which the included studies did not investigate. None of the primary sources explicitly assessed whether participants deliberately feigned symptoms, so we adopted the more neutral phrase symptom exaggeration to describe the findings without attributing conscious deception. We trust that this explanation provides a clearer rationale for our choice of terminology. We added the following material to the background:

“Also, terminology such as exaggeration, malingering, or over-reporting are defined inconsistently across studies, making it difficult to distinguish intentional deception from psychological amplification of distress.”

3. Both the introduction and discussion are very short, relating to the point 29 raised in the previous round. In my opinion, these comments are not sufficiently dealt with. Also the transitions between different topics within the Introduction and Discussion could be improved.

Reply 3: We expanded and revised our introduction and discussion sections and reviewed both to improve transitions.

4. It is stated that French and Spanish studies were included to reduce language bias, but what about culture bias? You state this later on, but this is something that could be taken into account at an earlier level.

Reply 4: Thank you for raising the issue of culture bias. Although we included studies in English, French, and Spanish to reduce language bias, we acknowledge that restricting our review to a North American setting does not fully address cultural variability. Indeed, cultural factors (e.g., varying attitudes toward disability) may influence how patients report symptoms or interact with examiners. This may have an effect on both the prevalence of symptom exaggeration and assessors’ interpretations. Our review did not find studies addressing this issue specifically within our inclusion criteria; therefore, we could not explore these possible differences. We agree that future research should investigate how cultural diversity affects IME outcomes, with attention to factors such as language barriers, health beliefs, and potential biases among both examinees and examiners. We have now clarified in the discussion section that this is a limitation of the included studies (and therefore our review), and highlighted exploration of the impact of cultural bias on IME findings for future research.

5. Risk of bias assessment: also relating to point 36> why were no tools used to assess risk of bias? E.g., Cochrane's? Also, what was the measure of agreement (e.g., Kappa)? The selection seems very subjective.

Reply 5: Studies eligible for our review were observational in design, and thus the Cochrane tool (designed for Randomized Controlled Trials) was not appropriate. Instead, we collaborated with content experts and methodologists to develop key criteria specific to known group designs (e.g., sample representativeness, reference standard confidence, and missing data). After piloting these criteria, reviewers underwent several rounds of training and independently and in duplicate assessed risk of bias, resolving disagreements by consensus or a third reviewer. Although we did not calculate a kappa statistic, we have clarified these steps in our Methods to illustrate how we increased reliability for our risk-of-bias judgments.

6. Table 2: what do the levels of inconsistency mean, especially 'serious inconsistency'? How does this relate to your conclusions?

Reply 6: In GRADE terminology, “inconsistency” refers to the variation in results across studies (ie, heterogeneity). When GRADE tables state “serious inconsistency,” it means there was substantial variability in effect estimates that we could not adequately explain (for example, no clear reasons for differing results emerged in our subgroup analyses). Unexplained between-study inconsistency lowers our confidence in the pooled estimate, which is reflected in our GRADE assessment of the certainty of evidence. We have expanded on this in a footnote under Table 2.

7. It is concluded that the review found no evidence for differences in the prevalence of symptom exaggeration based on clinical condition, but most patients among studies eligible for our review presented with either mild TBI or chronic pain'. This was not a (preregistered) research question, and how to conclude this based on only two clinical conditions? Other clinical diagnoses might have symptom exaggeration, but are not included in the current review.

Reply 7: We agree. Although our review did not detect differences in the prevalence of symptom exaggeration across clinical conditions, most included studies recruited individuals with mild TBI or chronic pain. Because few other clinical populations were studied, we cannot draw definitive conclusions for other diagnoses. We have included language in the results (prevalence of symptom exaggeration and additional analyses section) and strengths and limitations section of our review clarifying the populations that were represented and highlighting that there are other patient groups that were underrepresented or absent in our review and where more research is needed.

“We found no significant subgroup effects for type of clinical condition (mild TBI versus chronic pain versus other conditions)…”

“In terms of limitations, we restricted our review to IMEs conducted in North America and eligible studies focused mainly on chronic pain and TBI. The generalizability of our findings to other jurisdictions, contexts, and clinical conditions, is uncertain.”

8. Some reasoning errors in conclusions; if women report higher rates of symptom exaggeration, then it is logically that studies that included more women show higher raters.

Reply 8: We anticipated that studies eligible for our review would report a range regarding the prevalence of symptom exaggeration among individuals presenting for independent medical evaluations. Accordingly, we conducted several subgroup analyses to see if we could identify one or more factors that helped explain between-study variability. Only one of our subgroup analyses, the proportion of female participants, showed a credible subgroup effect. This result supported the reporting of separate rates of symptom exaggeration between men and women.

9. There are still some punctuation errors (extra comma use, delete space) throughout the manuscript. At times the phrasing (e.g., p.23 line 72 ('Such concerns...) could be improved.

Reply 9: Thank you. We have reviewed the manuscript more closely and corrected any errors.

10. For research transparency, what were the exact key words? I know they are stated in supplemental materials and that the apparently take 3 pages to explain, but I do believe these could be stated clearly in the manuscript.

Reply 10: We have added keywords used in our literature search to the manuscript, and it now reads as:

The search strategies were developed using a validation set of known relevant articles and included a combination of MeSH headings and free text key words, such as malinger* or litigation or litigant or "insufficient effort" and “independent medical examination” or “independent medical evaluation” or “disability” or “classification accuracy”.

Review 3 Comments

11. Dear authors of the article, I want to thank you for a well-prepared article.

The undoubted advantages of the article are: The text of the article is logically structured and understandable to the reader. The language meets the level and requirements of the scientific language. The article uses modern data analysis, adequate to the purpose of the study. The research is of high importance for science and practice. The authors openly and adequately note in the article limitations, implications for future research and practice.

I certainly recommend this article for publication in the journal, but I have some questions and recommendations for improvement.

Reply 11: Thank you. We appreciate you taking the time to review our manuscript and address your questions below.

12. Why were only these databases CINAHL, EMBASE, MEDLINE and PsycINFO used for analysis?

Reply 12: We selected these four databases (CINAHL, EMBASE, MEDLINE, and PsycINFO) because they collectively capture the key biomedical, nursing/allied health, and psychological/psychiatric literature most relevant to our review topic. Also, we evaluated, with the help of a senior medical librarian, a validation set of known eligible studies to confirm our search strategy and database coverage. When our search retrieved all studies in our validation set, we were reassured that our choice of databases and search approach was sufficiently comprehensive to capture the relevant literature in this area. We further supplemented our search by scanning the reference lists of included articles, and having experts review the list of included studies, providing additional checks for completeness.

13. In the discussion section, it is necessary to go beyond America and compare scientific data with other countries and continents

Reply 13: Thank you for your suggestion. We have now added to the discussion section data comparing prevalence of symptom exaggeration beyond America. The text reads as such:

“Although our review focused on IMEs in North America, data from other regions also suggest high rates of symptom exaggeration. An observational study in Spain reported that of 1,003 participants (61.5% female), drawn from unselected undergraduates, advanced psychology students, the general population, forensic psychologists, and forensic/legal medicine physicians, one-third reported having feigned symptoms or illness (44). Data from Germany and the Netherlands suggest that one‐fifth to one‐third of clients in forensic or insurance contexts exhibit symptom overreporting (45). Further, a Swiss study found that 28% to 34% of individuals undergoing medico‐legal evaluations demonstrated probable or definite symptom exaggeration (46).”

14. At the end of the paragraph "Eligible studies", clarify what does "large sample size" mean (what is the sample size in numbers)?

Reply 14: Regarding this sentence: “In cases where multiple studies had population overlap, we included only the study with the larger sample size.” We were concerned that some datasets may be used by multiple investigators, and the risk of double-counting the same patients in such instances. As such, we set a rule to avoid using data from the same patients more than once by excluding studies that reported on the same dataset. In order to pick a study to exclude (if this situation arose) we decided to exclude the study that used less data (the study with a smaller sample size). As this was a relative decision (in some cases, a smaller study could involve 200 individuals, and in others 1000 individuals) we did not pre-specify a sample size threshold. As it turned out, none of our studies had overlapping cohorts.

15. In the paragraph "We rated confidence in the reference standard as either..." describe in more detail (if possible, give examples) what ‘weak’, ‘moderate’ or ‘strong’ mean?

Reply 15: We have revised this section to provide more detail and examples. It now reads as:

“ We categorized the reference standard and rated our confidence in it as either : (i) ‘weak’ when the study declared a known-group design, however its only criterion for identifying symptom exaggeration was below-chance performance on forced-choice symptom validity testing without any corroborating clinical observations or inconsistencies in medical records. For example, a patient with a mild ankle sprain is labeled as exaggerating exclusively because they fail a below‐chance forced‐choice test of pain threshold, with no clinical exam or review of documented pain or functional abilities; (ii) ‘moderate’ where most patients exaggerating symptoms were identified by forced symptom validity testing results, but some cases could be confirmed using other credible indicators. For example, a claimant insists they cannot remember simple details of their daily routine (e.g., the route to their kitchen), yet is casually observed navigating complex tasks with no apparent cognitive difficulty; or (iii) ‘strong’ where exaggeration was determined by either forced symptom validity testing results or other credible clinical evidence. For example, a clinical finding that would classify a patient presenting with persistent post-concussive complaints after a very mild head injury as exaggerating symptoms would include claims of remote memory loss (e.g., loss of spelling ability).”

---

## [Decision Letter · Decision Letter 2]

Prevalence of Symptom Exaggeration Among North American Independent Medical Evaluation Examinees: A systematic review of observational studies

PONE-D-24-31136R2

Dear Dr. Busse,

We’re pleased to inform you that your manuscript has been judged scientifically suitable for publication and will be formally accepted for publication once it meets all outstanding technical requirements.

Kind regards,

Thiago P. Fernandes, PhD

Academic Editor

PLOS ONE

Additional Editor Comments (optional):

Reviewers' comments:

Reviewer's Responses to Questions

>**Comments to the Author**

1. If the authors have adequately addressed your comments raised in a previous round of review and you feel that this manuscript is now acceptable for publication, you may indicate that here to bypass the “Comments to the Author” section, enter your conflict of interest statement in the “Confidential to Editor” section, and submit your "Accept" recommendation.>

Reviewer #3: All comments have been addressed

>2. Is the manuscript technically sound, and do the data support the conclusions?

The manuscript must describe a technically sound piece of scientific research with data that supports the conclusions. Experiments must have been conducted rigorously, with appropriate controls, replication, and sample sizes. The conclusions must be drawn appropriately based on the data presented. >

Reviewer #3: Yes

>3. Has the statistical analysis been performed appropriately and rigorously? >

Reviewer #3: Yes

>4. Have the authors made all data underlying the findings in their manuscript fully available?

The PLOS Data policy requires authors to make all data underlying the findings described in their manuscript fully available without restriction, with rare exception (please refer to the Data Availability Statement in the manuscript PDF file). The data should be provided as part of the manuscript or its supporting information, or deposited to a public repository. For example, in addition to summary statistics, the data points behind means, medians and variance measures should be available. If there are restrictions on publicly sharing data—e.g. participant privacy or use of data from a third party—those must be specified.>

Reviewer #3: Yes

>5. Is the manuscript presented in an intelligible fashion and written in standard English?

PLOS ONE does not copyedit accepted manuscripts, so the language in submitted articles must be clear, correct, and unambiguous. Any typographical or grammatical errors should be corrected at revision, so please note any specific errors here.>

Reviewer #3: Yes

>6. Review Comments to the Author

Please use the space provided to explain your answers to the questions above. You may also include additional comments for the author, including concerns about dual publication, research ethics, or publication ethics. (Please upload your review as an attachment if it exceeds 20,000 characters)>

Reviewer #3: The authors correctly and competently provided answers to all questions and comments, specifying in detail the corrections that were made to the text. The authors have done a lot of work in the process of preparing the article. The article turned out to be interesting for the professional community and has theoretical and practical significance.The authors openly and adequately note in the article limitations, implications for future research and practice. I certainly recommend this article for publication in the journal

>7. PLOS authors have the option to publish the peer review history of their article (what does this mean? ). If published, this will include your full peer review and any attached files.

**Do you want your identity to be public for this peer review?** For information about this choice, including consent withdrawal, please see our Privacy Policy .>

Reviewer #3: **Yes: ** Olga Terekhina

---

## [Editor Report · Acceptance letter]

PONE-D-24-31136R2

PLOS ONE

Dear Dr. Busse,

I'm pleased to inform you that your manuscript has been deemed suitable for publication in PLOS ONE. Congratulations! Your manuscript is now being handed over to our production team.

Kind regards,

on behalf of

Dr. Thiago P. Fernandes

Academic Editor

PLOS ONE